# Nigerian Migrant Women and Human Trafficking Narratives: Stereotypes, Stigma and Ethnographic Knowledge

Estefanía Acién González [1,2] 

1    Department of Geography, History and Humanities, University of Almería, 04120 Almería, Spain; eacien@ual.es
2    Center for the Study of Migrations and Intercultural Relations (CEMyRI), University of Almería, 04120 Almería, Spain

**Abstract:** During the last decades, Nigerian migrant women in the European sex market, described as victims of trafficking, have generated consistent concern and outrage. This article analyzes data from an ethnographic study of more than 800 Nigerian sex workers in southern Spain, describing the networks used by these women to carry out their migration projects and the relationships they establish with their agents. Thus, it contributes to refuting the hegemonic narrative about trafficking and its victims by contrasting it with data collected and systematized over almost a decade of participant observation and informal conversation. This paper argues that the stereotypical image of the Nigerian migrant women as victims of abuse and violence by transnational trafficking networks functions to justify strict migration-control policies and the denial of labor rights to sex workers. As an antidote to the dominance of narratives based on stereotypes and pseudoscientific claims, this paper underscores the urgent need for ethnographic research and its focus on *emic* (participant) perspectives. The goal is to develop tailored and effective policies and practices for the prevention of and intervention in migrant women's experience of exploitation, abuse, and violence.

**Keywords:** human trafficking; trafficking networks; women migrants; sex work; ethnography

## 1. Introduction

The victim of trafficking has traditionally been described as a woman, poor and alone, obliged to submit to "forced prostitution" and, above all, over the last decades, a migrant from an impoverished country, unscrupulously managed by criminals from whom she can only hope to be rescued by a "developed" state (Agustín 2004; Kempadoo et al. 2012). This image is shaped by socio-cultural reality (Cunningham and Cromer 2016), as is clearly perceptible in the case of the Nigerian woman in Europe, where the anti-trafficking narrative shows a victim of atrocities, deception, manipulation, extreme control, sexual aggression, exotic extortion through voodoo, and the obligation to pay large amounts of money to sophisticated, tightly organized, and cruel networks in order to free herself from trafficking (Kastner 2010; Baarda 2016).

A brief search on the web shows dozens of headlines such as: "Threats, voodoo rituals and human trafficking: four accused in Malaga for sexual exploitation" (El Periódico de España, 3 April 2023) or "Voodoo, the great threat to Nigerian women victims of trafficking" (La Vanguardia, 7 June 2016). The repetition of these messages in the press correlates with recent interest and increasing resources for the fight against trafficking, sustaining an image of Nigerian sex workers as victims of a hell caused by heartless, exploitative Nigerian traffickers.

Based on an ethnographic study of more than 800 Nigerian women, conducted between 2005 and 2012 in the sex market of the Poniente Almeriense (southern Spain), this article shows the inadequacy of these homogenizing discourses and their stereotyped picture of Nigerian women who migrated to Europe. The testimonies of these women

unsettle that image in their stories about the structure of migratory networks and their relationships with the people who comprise them. The research shows a wide spectrum of migration experiences, attitudes, and difficulties that embody intersectionality to an extent that can be difficult for Europeans to assimilate without the help of stigmas and stereotypes (Acién 2019). Many of the women I knew had traveled with persons who profited from taking advantage of their need, but to see only a group of victims with little capacity to make choices has erased them as active subjects. Thus, I focused on their own descriptions of their migration experience to see how well the hegemonic narrative matched reality. The results obligate us to critically review the existing literature and political rhetoric, putting the subjects' experiences at the center, because these protagonists not only experience or suffer their circumstances but they also interpret them. It is important to know their actual stories, analyze them and take them into account in order to comprehend the complexity of the migration phenomenon and throw light on the cracks in the anti-trafficking system.

I will first show how the hegemonic anti-trafficking narrative presents a biased image of the living reality of migrants and spreads a stereotypical story of the victim that justifies restrictive migration policies and the repression of prostitution. Secondly, I will present the testimonies of Nigerian women to show the inadequacy of this narrative and the enormous diversity and complexity with which they organize their journeys. To this end, I will discuss forms of financing and the relationships entered into with agents specialized in organizing irregular journeys.

*The Anti-Trafficking Narrative*

A historical characteristic of the trafficking story is that it makes the trafficking networks solely responsible for the living conditions of migrants in sex markets, naturalizing the effectiveness of punitivism and the need to extract testimonies of rescued women. However, more and more specialists warn of the narrow-mindedness of this vision and the strategy it justifies, which ignore the complexity of decisions, links, and experiences of those who migrate, as well as the context of cultural and structural violence in which their migration takes place—inequalities and imbalances, sexism, racism, migration politics, and the configuration of labor markets (Juliano 2002; Bernstein 2007; Kempadoo 2015; Acién and Checa 2020).

The dominant narrative reproduces the idea of a vulnerable and passive sex slave, exploited by a sexual predator, and needing to be rescued by the State (Torres 2016; O'Brien 2019). It manipulates the expression "taking advantage of the victim's vulnerabile situation" to dismiss poor women's capacity for social agency, criminalizing a multitude of gray behaviors, many of which do not violate the autonomy of the alleged victims (Sánchez 2022). This is how the abolitionist ideological perspective (Lamas 2016) ignores the consent of those who offer sexual services, ideologically negating their sexual freedoms and autonomy and victimizing their sexuality (Maqueda 2020). Using the term *victim* aims to stir emotions in a society that still considers women's sexuality to be essentially linked to affection, generosity, and attachment. In that view, a woman who cannot match the ideal is practicing an unnatural, unpleasant, dangerous act, out of place everywhere, a consequence of alienation, immorality, or others' violence.

As we see with the concept of trafficking, the status of victim is legally defined as a person who "meets the requirements stipulated for victims in the legal norms and protocols that determine it" (Meneses 2016, p. 22). Trafficking is not an objective and universal social phenomenon, but a crime, a juridical concept that is objectified by protocols and normative codes. The reality is much more complex, and it is the social majority, through the police and, ultimately, the judiciary, who place the label of trafficking on what happened or the label of trafficking victim on a person's experience. This is why there are standardized indicators,[1] consistent with written norms, which serve to identify victims among the populations considered vulnerable to the crime. It is precisely the legal definition of trafficking and the content of these indicators for detecting victims that imprint the static

and stereotyped image that was so difficult to recognize among the hundreds of Nigerian women working in the sex market that I met and with whom I lived.

Several researchers have called into question the legalistic priorities of the anti-trafficking system, which seem designed more to justify and reinforce the immigration controls against poor countries and eradicate prostitution than to rescue and restore victims (Anderson and Andrijasevic 2008; Kempadoo 2015; Piscitelli 2016; Clemente 2017). For decades, other researchers have criticized those who design migration control policies for their use of anti-trafficking rhetoric (Maqueda 2000; Mendoza 2000; Wijers 2001; Nicolás 2007; Goździak 2021). In 2005, the European Court of Human Rights (ECHR) urged states to address the structural causes that favor trafficking, alerting them that migration policies could be feeding the problem, as well as violating fundamental human rights in their implementation (Classen and Polanía 1998; Juliano 2002; Mestre 2011).

In addition, I should mention a growing concern that these punitive policies have been naturalized in recent decades as the only solution to violence against women. The fact that gender violence has occupied an important place in the political agenda, in line with feminist demands and as represented in movements such as "ni una menos" (not one less),[2] "me too"[3], or "se acabó" (it is finished),[4] has gone hand in hand with stiffer penalties for abusers and aggressors, as well as sophisticated tools for the prosecution of crimes and the expansion of their definition (Daich and Varela 2020). More and more voices are warning about the perverse effects of stiffer penalties on the fundamental rights of women in general and the victims of these crimes in particular. They criticize the tendency to amalgamate microaggressions and continuous violence, viewing all of them as part of one continuum in regard to criminal punishment, and so all is resolved by measuring the seriousness of the crime and deciding the amount of the fine, the disciplinary measure, or the number of years in prison. Critics of this logic affirm that the logic of "everything is a crime" does not "help focus protection measures on the gravest cases or implement the necessary resources for people in conditions of great vulnerability, and [public policies should] consider these women as subjects with decision-making capacity and not objects of intervention" (Caro and González 2023), given that punitivism does not allow victims a voice in criminal proceedings that were set in motion when they reported the problem. For all these reasons, it is common to find proposals for alternatives such as restorative justice (Villacampa 2020).

This debate within feminism concerns the effects of a punitive anti-trafficking system that is not designed "to understand that the protection of the victim is a value in itself and a purpose of the procedural action" (Guil and Rodríguez 2021, p. 7), since its univocal priority is to prosecute a crime. Therefore, despite the international recognition of the rights of victims of trafficking,[5] there are constant accusations of victimocentric approaches, on one side, and deeper criticism of the security system. In the name of feminism and the safeguarding of women's rights, these critics say that the prosecution system endangers the welfare of those identified as potential victims of trafficking not only with its victimocentric approach but also, and more seriously, by fostering repression in their work and relational environments (Bernstein 2018). This approach feeds a system of intervention on which organized civil society entities subsist, and it strengthens the bureaucratic apparatus of the state (Clemente 2022; Sierra and Clemente 2023).

On the other side, the policies of prostitution abolitionists displace concerns for trafficking in other labor sectors in favor of hypervigilance on sex work (and its clients) (Bessa and Silva 2019). This results in increased police pressure on those who work in prostitution, generating fear and a loss of trust in the system (especially in the police). This paradoxical effect—prostitutes fear those who are supposed to save them—is a direct violent consequence of the narrative that prostitution should not exist, and that it is the consumption of sexual services that fuels trafficking (and not unequal structures or the difficulties of migration). These claims are largely pseudoscientific and ideological, but they support a multitude of public policies that impact the daily lives of sex workers (Acién and Checa 2020).

Finally, it should be added that certain feminists' projection of the stereotypical image of migrant women in the sex market as victims of trafficking makes real women invisible and invalidates their voices. This same feminism, the one most frequently present in power (Uría 2012), usually holds that we have never been more in danger, especially those who are supposedly subjected to patriarchal institutions such as prostitution (Osborne 2007, 2010), that is, the same women who are considered victims in need of rescue by a state, which, in addition, must imprison the culprit. Thus, as I said above, punitive policies, promoted as feminist, silence and discredit the stories of the protagonists, making anti-trafficking policies a space for controlling migrant women instead of a tool for the emancipation and support of their freedoms, rights, and the development of their personal projects (Clemente 2023).

In this work, I seek to counteract this phenomenon, amplifying the voices of female sex workers who are seen as victims of trafficking, and listening to what they have to say about their journeys and projects.

## 2. About the Method

This paper proposes to offer a description of a part of the migratory experiences of Nigerian women in Almeria based on an ethnographic study that took place between 2005 and 2012 among more than 800 Nigerian women who sell sex in the Poniente Almeriense, in the south of Spain. I have selected the most representative testimonies about the organization of the journey and the relationship established with the different agents of the trafficking networks. These accounts were extracted from informal conversations, thanks to the ethnographic opportunity provided by my work as a health mediator for the Association for Human Rights of Andalusia.[6]

Almost all my informants were under 30 years of age, and 75% of them were of *Edo* ethnicity. Most came from an urban environment, 70% from Benin City, where their parents migrated after losing their traditional way of life. The vast majority of them were in Spain in administrative limbo and worked as sex workers, mostly in dwellings located between agricultural enterprises where they could contribute to the family economy. A total of 64% began their journey when they were over 20 years old, and the rest as adolescents. The most frequent journey trajectory lasted from one to four years, crossing the countries separating Nigeria and the Strait of Gibraltar in a straight line and staying in Morocco for months or years awaiting a boat to the south of Spain. About 37% had contracted debt to cover travel expenses, 42% had traveled with their own or family resources, and about 21% had received altruistic help. The ethnographic accounts, however, document that most were pressured by organizers of irregular travel. Even so, they describe experiences of complexity and diversity, demonstrating their agency and their attention to symbolic and discursive representations[7].

Ethnography proved to be the epistemological strategy most compatible with my role as a mediator in the field. Its methodology guides the researcher toward a holistic grasp of the object of study and the context in which it was located.

The Poniente Almeriense is home to a strongly gendered and ethnified labor market[8] (Reigada et al. 2017), where intensive agricultural exploitation under plastic provides the majority of jobs for male migrant wage earners, while some women work gendered tasks at warehouses and in the service sector, especially care-giving, cleaning, and hospitality. The women's national origin is significant. My ethnographic work showed that Nigerian women were not able to access these labor niches to the same extent as other migrant women, so that sex work ended up being one of the few options for survival (Acién 2015).

Ethnography applies tools for collecting data in a non-invasive way through participant observation and informal conversation, which is ideal in stigmatized contexts. It is able to reveal the subjectivity of the subjects, their knowledge, their potential for interpretation of how migration and anti-trafficking policies impact their lives, and the motivations behind their decisions, projects, and desires.

The selection of testimonies covers two thematic blocks: (1) strategies for financing their migration and (2) their relationships and experiences with the organizers of their journeys.

## 3. Results

### 3.1. Financing Strategies

The women I worked with used two basic strategies for financing migration journeys: (1) use of their own resources, family, and even bank debt, and (2) indebtedness to be paid off at the destination. In the latter group, all agreed that selling sex was the only possible way to raise the money needed to repay their debt, given the circumstances of the local labor market. Only a small number saw themselves as forced by third parties with threats and loss of liberty.

It was common for them to seek contacts at the point of origin to plan the journey and to know that sex work would be part of the process upon arrival in Europe. Many of them used the term *family money* to note that their relatives took part in the migration project and understood the diaspora context, which reinforces the thesis of a new economics of migration (Stark and Levhari 1982; Stark and Taylor 1994):

> Girls who go to Europe know this is the job. I knew it. But I told my mother I was coming to study. All my brothers are abroad, in Italy and America. My parents gave me the money to come. We are all abroad. If my mother found out, she would kill me. I was asking who I could come with and I found him. My mother didn't know anything. (Happy,[9] September 2003)

Planning ahead, the family may even take loans from financial institutions. Many stories show awareness of the final amount of their debts, but not the difficulties they will have raising the money at their destination.

> I called a woman who told me that the trip was going to cost me more than six million naira [some 26,300 euro]. I went to the bank and asked for the money, with my father. I have already paid for my trip, now I pay the bank in Nigeria six and a half million naira [about 28,500 euro]. I pay every month 300 euros to the bank, but it's 1000 euro a month. The rest is paid by my father. (Sandra, June 2005)

Occasionally, the family provides support along the way, according to its ability and in response to needs at each stage of the trip. Although this is not common, it is interesting, since it helps us see that it is a much less expensive trip for those who have resources than for those who rely completely on debt:

> I had money and then my father would send me [money] on the road when I had no more money. I didn't spend 30,000 euros like other girls, no! I don't know how much I spent, but when the money is yours, it's not that much. (Helen, August 2004)

There are also experiences that combined payment commitments and spaces of autonomy during transit, thus reducing the debt. Some testimonies showed the attainment of resources during the itinerary or the wait in Morocco[10]. Selling sexual services or asking for alms—they call it *to* do Salam Alaykum—are common strategies in these cases. However, having one's own resources during transit can pose risks of abuse and theft.

> Working! I pay for everything by working. He earns little money, but pays everything, little by little. Work in Africa, on the road, in Morocco, Roquetas. I pay for the trip with my body, everything, everything. (Florence, November 2006)

> I saved my money twice to go through Nador, but they stole it from me both times! I did everything: *salam aleykum*, I asked my sisters for money, I sold food, I did things that my companions asked me to do. . . Everything I could! (Destiny, October 2006)

Trying to maintain autonomy on the road is complicated, but many try with varying degrees of success. The following testimony is especially valuable because it shows the agency's ability to avoid total control by travel organizers, trying not to mortgage one's freedom:

> In many places he asks for money on the street to pay for my food, my clothes, Orange card. He asks in the street, from the people. In Mauritania, Algeria and Morocco. Many girls ask everything to their *employer*. I don't want to ask for everything, if he asks for everything, you give everything. Always hide your money. They ask: "Do you have money? How do you pay your phone?" Ah, very difficult! Ah! (Mary, October 2004)

When fully indebted, the accounts reveal that the vast majority of the women were not forced into prostitution. The pressure exerted on the women is related to debt repayment. Those who put pressure on them are certainly aware that in Europe they cannot easily access other jobs. In other words, the *trafficking networks* take advantage of the situation of exclusion to which a segmented, gendered, and racist labor market, combined with migration policies, leads these women, who are newcomers, black, migrant and irregular. In this context, working as a sex worker is the only way to raise money.

I will now turn to a description of the different organizing agents that I have detected and the relationships that women establish with them, as well as the nature of the pacts and relationships that are established.

### 3.2. A Smuggling Agents: Variety, Complexity and Ambiguity

The UNODC (2015) recognizes that, although smuggling of migrants involves professionalized criminals, not all the agents involved have this profile. There is plurality depending on various factors ranging from their origin and cultural diversity to stages of transit, demographic and socio-cultural characteristics of those who travel on their behalf, the size of the groups they manage, etc. I detected complex situations that were difficult to label, including variety in the levels of professionalization and stability of these agents, in their capacity to exercise power and violence, in their abilities to trace successful, uninterrupted itineraries and to adequately fulfill their assignment, and even in the levels of cohesion of a supposed network that was sometimes described as compact, but more often than not disorganized, variable, and improvised, although finally someone, usually a woman at the destination, picked up the baton with toughness and ensured the final profit. I will deal with all of this below.

One of the most relevant issues to take into account when it comes to understanding the high level of commitment to and vital commitment to the migration project is the total assumption of the harshness and dangers of the journey and even the fact that prostitution would be practiced once in Europe:

> Can I tell you the truth? If I tell you the truth, I didn't care, I knew it. My mother was crying, because I could die in the sea and she told me not to come, too much suffering. My mother told me to wait until she saved money to travel with her money. But I didn't want to wait, so I looked for someone to help me and I paid. (Blessing, May 2005)

However, it is also common to find testimonies permeated with frustration from those who have been—and have felt—deceived both about the access to job opportunities or for having underestimated the difficulties of debt repayment.

> The woman I talked to in Nigeria about coming to Europe told me I would have a good job. And look at the job. Well, huh [with sad irony]. I can't talk about this with my mother. When I told her, she got sick. (Joy, November 2004)

There are times when the planning is solid, but, at other times, the women make contact with seemingly unconnected people during the different stages. Thus, the itinerary may be perfectly organized from the beginning and then vary due to political incidents

(changes in supra-state or state decisions on border control) and ecological or various unexpected events such as health problems, personal conflicts, aggressions, detentions, deportations, etc. All this makes it impossible to draw a unanimous and generalized transit experience.

Originally, the recruitment of female travelers on the streets of Benin City is a well-known phenomenon, but, as I said, it is usually the women themselves who seek help to travel and do so through close contacts, sometimes even relatives, neighbors, or friends; so it is difficult to establish the exact moment at which the price is set or the nature of the relationships that develop, whether economic, dependent, or even affective. In short, however compact a trafficking network may appear to be, there are a multiplicity of factors that affect the final experience.

According to my data, the roles played by each of the different organizing agents develop with a certain spontaneity, becoming institutionalized and forging more or less co-ercive styles or dynamics as patterns and strategies for adapting to the challenges presented by travel are repeated. Thus, those who take on a specific function to facilitate transit end up having as much power over travelers as they can exclusively possess through certain social and informational capital beyond the reach of the people who need them to travel. As we shall see, the way in which each person expresses this power and what they use it for is another matter, and in this sense, women express a variety of situations in terms of levels of coercion and violence.

In this sense, it is important to relativize the story presented by compact and profes-sionalized smuggling networks. It is just as important to denounce their abuses as it is to pay attention to the variety of situations, since the activities of all the agents involved cannot be measured with the same yardstick (De Haas 2007, 2008). Some actors are part of criminal networks, wield power, and use violent methods, but others engage in more innocuous, less lucrative activities carried out in order to survive or subsidize part of their own way (Women's Link Worldwide 2014).

Here, I speak of guides, connection men, patrons, and madams, which are the roles I learned about through the women's descriptions.

The so-called UNODC (2011) smugglers, who perform the task of what I call guide men, offer their services as specialists, especially, as Orji (2003) describes, when the journey approaches the desert and requires specialized vehicles and safe routes that the guides agree to provide for a fee. A guide man is the person who points the way and tells what to do. He is a specialist in transiting unknown and dangerous territories. Throughout the itinerary, women are likely to encounter a multiplicity of agents who can be recognized under this label, which, in reality, has a rather elastic content, since their functions depend on the needs imposed by the context or their particular skills.

Women share the road with men, and guide men serve the same function for all travelers. However, it seems that abusive or sexual treatment of women is more frequent. I collected several testimonies in this direction, but the experiences were ambivalent and diverse. There were those who expressed good experiences and a positive view, those who reported having had affectionate relations with some of them, those who consented to sexual relations seeking facilities, and those who denounced sexual abuse.

> A lot of kids fall by the wayside and are guide men. They make a living. I had no problem with them. A lot of them are good people, they help you hide, they explain things to you. (Kelly, June 2005)

> A lot of girls fuck with guide man because they don't have money for food. They take care, give food, everything. (Susan, February 2007)

> I starts to fuck on the trip. A lot of girls look for a boyfriend on the trip to survive. If you have a boyfriend, nobody touches you. My boyfriend was a *guide man*. (Joy, February 2005)

> Men who carry people fuck girls. She says that if not, she will die because she will not eat and without a boyfriend she cannot survive. A lot of pregnant girls have their child in the desert, no doctor, nothing, nothing. (Costas, March 2008)

Another figure widely represented in the testimonies I collected was the *connection man* or *woman*, which the UNODC (2011) points out as the most powerful and most profitable in the commercial chain, while, in my field notebook, a migrant in transit appears, who has established herself at a key point of the itinerary such as Tamanrasset, Casablanca, Rabat, or Tangier (sometimes because she has not been able to move on); who has valuable experiential and relational baggage whose sale to the other migrants in transit and other agents dedicated to migrants in transit and other agents dedicated to the sale of their products, Casablanca, Rabat, or Tangier (sometimes because they have not been able to move on); and who has a valuable experiential and relational background whose sale to other migrants in transit and to other smuggling agents gives them a privileged status and the possibility of earning a living with varying degrees of success. Thus, it is likely that their status depended on their successes, seniority, and strategic ability to relate to other powerful agents or local authorities of time at the same point in the transit journey, such as document forgers, transporters, and so on. Here, we see an example of verticalization in migrant relations, when "information and contacts become an economic value in the hands of a few" (Pedone 2010, p. 104).

In Morocco, I met a woman and a man whom my informants identified as a *connection*. The valuation of these figures in the stories depends on the human quality with which they played their role.

> Johnson connection is a good man. He helps girls a lot. Connections are necessary and he is a very good person. Others are not. Others take advantage of the girls, because they don't have power. (Sweet, August 2007)

> There are connections that are good people and help the women a lot with their children, calling the doctor, looking for help to go to Europe for less money. Others are much worse because they take advantage of the girls, they cheat them. (Mary, October 2006)

In the case of the employers, the reference was to powerful agents in transit, while the madams are already at their destination. Both agents appear to be at the top of the hierarchy, and from the way they operate and the type of control they exercise, especially the madams—who ultimately collect the debt—it can be deduced whether or not a woman's experience is close to the legal content of being a victim of trafficking for forced prostitution.

For Women's Link Worldwide (2014, p. 28), the patron is "the person in charge of the control of women and adolescents in the countries of stay in North Africa, on behalf of the madame, as well as negotiating the connection to go to Europe in the moments prior to the crossing". My perception partially coincides with that of these researchers insofar as it would explain their legitimacy to exercise control and their absence in Spanish territory. However, it is not so clear that the boss is under the orders of the madame, unless she controls the process from the beginning, or at least from the arrival of the women in Morocco. My data attribute to the patron only the ability to negotiate a price and transfer the travel debt to another person.

> When the girl is in Morocco she has to go to Europe! The boss calls someone and sells you. He says "Do you want a girl for Madrid?" He says what your name is, how old you are, if you have children. . . everything. And he sells the girl to a madam. (Faith, November 2008)

As in the previous cases, the testimonies give ambiguous messages about the moral quality of these people. I also found references to the establishment of sexual relations, consensual or not:

> The boss takes care of you. He accompanies you taking care of you and there are some who take good care of you and others who rape the girls. Many girls

become girlfriends of the boss. If he is your boyfriend, he takes better care of you. (Mariama, January 2007)

He's your boss! If you don't do everything he tells you to do, he beats you! (Joy, March 2009)

My employer was very good. He protected me from the police and gave me everything I needed. I looked at other girls who were bad and I wasn't so bad. (Queen, October 2005)

What seems clear is that the employer, when he or she exists, is responsible for the protection, control and monitoring of a girl who is waiting to be introduced to a madam who will receive the final payment for the trip. What is not so evident is that the exchange has been made or stipulated beforehand, nor that it comes pre-designed from the beginning of the trip.

In some research, it is claimed that the madam is literally in charge of "the exploitation of the victim at the destination" (Women's Link Worldwide 2014, p. 28) and that she is the one who controls the process of the trip from beginning to end (UNODC 2011). However, I did not find evidence of their presence from the beginning of the itinerary, but of their appearance at destination or shortly before. In addition, I did not find that the women were directly forced into prostitution, although they were forced to pay the travel debt, exercising control until it was paid off. That is why the women worked where the madams told them to if they could not find other alternatives.

If the madam is a good person, she charges little and just wants to get paid every month. That's the way it is. If she is a bad woman, she beats you, she asks for more money, always more money. She just wants money and if she wants money fast, she controls you all day long [...] she wants you to work as a whore to get paid soon. (Christiana, June 2008)

My madame is a very nice person. She has always treated the girls very well. She charges little and doesn't look at your cell phone. You just pay your money and she leaves you alone. (Joy, January 2011)

My madame is in Valencia. Now she controls me a lot. I can change jobs, but I have to tell her because I have to pay her every month [...]. But she is a good person. She introduced me to everyone here. You can't do anything if you don't know people. (Queen, November 2004)

Illustrating the fact that the pressure is on to collect the debt and not so much to sexually exploit the women, on several occasions, I heard that the madam didn't care how they earned the money. They could leave prostitution as long as they kept paying. The problem was that no activity they could do, without a residence and work permit, was profitable enough to be able to pay such an amount of money in an acceptable time frame and that would not mortgage the best years of their lives. That is why paying off the debt quickly was a priority over giving up prostitution.

As a whore it is easier, you earn more money and there is always work. Last year I had a boyfriend and he helped me pay my madame every month. I worked as a hairdresser. But as a whore you earn more and I want to pay soon. I want to get it over with now. (Josephine, March 2008)

In the following case, there is a testimony where the madam was related to an informant and linked to her by an intense relationship of control–affection, which increased the ambivalence in the valuation. This circumstance entailed certain advantages, such as flexibility in the payment of the debt and a close and familiar relationship. On the other hand, the kinship and the differences in age and status implied domestic obligations for the girl, who complained of spending much of her time attending to his demands.

She takes care of my daughter when I'm working, but that's why I have to do a lot of things at home. But I have to do it, she helps me, she is my older cousin.

> That's why I pay less than other girls and if I don't pay for a month, nothing happens. But I have to work for her. That's the way things are in Africa. (Lauryn, October 2006)

I captured situations of forced prostitution and extreme control if the informants referred to pressures to work in specific places or threats if they tried to change their occupation or location. In the former, even under control, the woman was able to move to where she believed she was less at risk. In the second, a total lack of freedom was expressed.

> I have two bosses. One is in Madrid, but the most important one is in prison in Mali. She will be there for five years. I have to pay her. The one in Madrid controlled my work in Madrid. I wanted to work in a club, but she didn't want me to be on the street. She wanted me to be on the street. It's dangerous there! They killed two friends and threw them in the garbage, and my madame didn't care. She said, "It's life. That's why I came to Almeria. Here there are more Nigerians and I don't work on the street. But I have to pay her every month. If not, she will look for my family, for sure! I want to work and get rid of this problem. I want to pay soon. (Joy, November 2004)

> We are under control here. She's mean, she beats the girls. She keeps the papers and asks for all the money. We work and she takes the money. Then she pays for our food, our clothes, the phone, the cab. Everything. (Becky, June 2004)

## 4. Discussion and Conclusions

Not all Nigerian women financed their migration in the same way, as some sought resources through their families or by getting into debt with a financial institution. In addition, among my informants, many knew that they would engage in selling sex once at their destination, accepted the conditions of the pact, and, in fact, considered that they had to comply with it. Moreover, it is not unimportant that, if they felt that the madam was not complying with the stipulations, they were entitled to denounce the abuse without expecting reprisals.

In this sense, the reality of "smuggling" from Nigeria and how it is rooted in a socio-political reality impregnated with violence and corruption is widely described, as well as the confirmation of the origins of the problem in poverty and the lack of public protection against it, leading to the search for submerged strategies to survive (Barimah 2013). Therefore, we cannot lose sight of the fact that women swim in these waters, seeking their future trying to breathe in an intoxicated society and a suffocating international context that does everything possible to prevent them from getting out of it.

There are no homogeneous profiles among the protagonists of this work, just as there are no homogeneous profiles among those we call travel agents. That is why I insist on pointing out that neither the former respond to certain stereotyped images of trafficking victims nor do all the pieces of the machinery that we call smuggling and trafficking networks respond to the image of the ruthless criminal pursued by the anti-trafficking system.

Generalization leads to the victimization of all migrants and criminalizes their environments, making their strategies for developing their migratory projects invisible and multiplying the stigma they already bear. The discourse focuses only on the phenomenon of trafficking without taking into account the diversity of situations and experiences, which undermines any possibility of seriously addressing the most serious situations of control and exploitation and the structural causes that fuel and exacerbate them. Carling (2005a, 2005b) points out, among these, the corruption and political, ethnic, and religious violence that turn Nigeria into a country with no future from which women must escape[11] by resorting to networks that know how to circumvent European anti-migration legislation.

From my point of view, it is necessary to look beyond trafficking and not ignore different situations for this group of women. My work shows, for example, more situations of smuggling—very coercive and dangerous—than of trafficking and forced prostitution. For this reason, I believe it is important to pay attention to the complexity and diversity

of the realities exposed, and all this can only be documented and explained by applying an ethnographic approach, prioritizing the *emic* approach (Kastner 2010), and making the discourse of the protagonists prevail.

Current European migration policies prevent millions of people from traveling regularly, who resort to trafficking and are subjected to abusive scenarios where women become an opportunity for profit in the sex market, especially when it is the only place in the labor market where travel debts can be paid. The difficulties imposed by migration policy to access stable residence and work permits (and full citizenship rights); a segmented, gendered, and ethnified labor market; public policies hostile to prostitution; and the punitive anti-trafficking system (Plambech 2014) make up the pieces of a machinery that keeps these women in the most terrible economic precariousness, job insecurity, vulnerability to abuse and violence, exposure to police pressure, and hypervigilance of their lives and that of their close environments (children, friends, family, and anyone who approaches them), all contaminated by the puta stigma (Pheterson 2023; Sánchez 2023). Of course, if they manage to escape from these situations, it is because they develop complex strategies that are difficult to understand for the well-informed citizenry (Juliano 2006) and they pay a high price. In my opinion, the strength, capacity for resistance, and resilience that they demonstrate on their path to social integration distances them far from the stereotype of victim.

Therefore, I consider it important to address, describe, and interpret women's migration projects and the factors that condition them from the social sciences. To this end, it is necessary to prioritize the narratives issued by the subjects and place them in the holistic complexity in which such narratives and the realities they describe take place. Pointing only to coercive experiences, emphasizing only the lurid and trafficking situations, and placing Nigerian women exclusively in the role of victims perpetuates a moral position against prostitution per se and biases the knowledge of the phenomenon at hand (Peano 2013). This only manages to decontextualize reality and hinder its scientific analysis. Insisting on affirming and reiterating that they are mere victims of trafficking only contributes to increase their social stigma, but not to a better understanding of what is happening.

**Funding:** This research received no external funding.

**Institutional Review Board Statement:** The study was conducted in accordance with the Declaration of Helsinki, and approved by the Institutional Review Board (or Ethics Committee) of UNIVERSITY OF ALMERÍA (protocol code Ref: UALBIO2023/022 and date of approval 18 July 2023).

**Informed Consent Statement:** Informed consent was obtained from all subjects involved in the study.

**Data Availability Statement:** The data presented in this work belong to the set of ethnographic data collected for the preparation of the PhD thesis "Nigerian Sex Workers in Poiniente Almeríense": https://dialnet.unirioja.es/servlet/tesis?codigo=111594 (accessed on 13 January 2024).

**Conflicts of Interest:** The author declares no conflict of interest.

## Notes

1. In this regard, the document that compiles the indicators of trafficking agreed upon at the international level is the one published by the United Nations Office on Drugs and Crime. See https://www.unodc.org/documents/human-trafficking/HT_indicators_S_LOWRES.pdf (accessed on 13 January 2024).
2. https://niunamenos.org.ar/ (accessed on 4 December 2023).
3. https://metoomvmt.org/ (accessed on 13 January 2024).
4. https://elpais.com/sociedad/2023-09-03/se-acabo-el-pico-que-revento-el-sistema.html; https://elpais.com/opinion/2023-08-26/se-acabo.html (accessed on 28 March 2024).
5. As in Directive 2011/36/EU of the European Parliament and of the Council of 5 April 2011, https://www.boe.es/doue/2011/101/L00001-00011.pdf (accessed on 28 March 2024).
6. www.apdha.org (accessed on 10 January 2024).
7. Details of the total ethnographic study data can be found in the full version of the research by following this link https://www.educacion.gob.es/teseo/imprimirFicheroTesis.do?idFichero=CbCsTSdjdgw%3D (accessed on 28 March 2024).

8    In the province of Almería, women account for 43.46% of the active population, 41% of Social Security affiliation and only 31% of those affiliated with the Special Agrarian Regime. However, they are 91.76% of those hired for domestic service (Servicio Público de Empleo Estatal 2022). On the other hand, 81.02% of contracts in agriculture were made to foreign men (Servicio Público de Empleo Estatal 2023).

9    To ensure the confidentiality of the informant, the names shown here are not real.

10    The reality of those who manage to earn some money through these small strategies has also been reported by the UNODC (2011), which describes how migrants stay for long periods of time in places where they manage to earn some money in temporary jobs to obtain resources. These jobs or opportunities arise from contacts established in businesses (bars, inns, etc.), generally set up by women, also migrants, to meet the needs of travelers.

11    Women's Link Worldwide (2014) also spoke of the deterioration of traditional ways of life as a result of colonization processes, the idealization of ways of life in European countries, a discriminatory gender structure for women within the ethno-cultural contexts of origin (in particular, *Edo*), the sexual violence towards women that results from it, and the normalization of trafficking in their contexts of origin.

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
