# Peer review of "Nigerian Migrant Women and Human Trafficking Narratives: Stereotypes, Stigma and Ethnographic Knowledge"

_socsci, doi:10.3390/socsci13040207_

Round 1

Reviewer 1 Report

Comments and Suggestions for Authors

The article entitled “Nigerian Migrant Women confront Human Trafficking Narratives. Stereotypes, Stigma and Ethnographic Knowledge” aims to deconstruct some discourses around human trafficking that are occasionally used to abolish prostitution and justify repressive migration policies. In doing so, the article uses data collected during extensive ethnographic research in Spain with Nigerian women.

The article's argument is clear and the proposal certainly constitutes an interesting contribution to the empirical research on the topic.

However, I recommend revising the following items:

-    The current title makes one think of a research article on the ways in which Nigerian migrant women confront trafficking narratives, but this is not the aim of the article. Therefore, I suggest renaming it. Replacing the verb “confront” with the conjunction “and” might be sufficient.

-    In the section “The anti-trafficking narrative”, the article uses a body of work that has highlighted the controversial policies omitted from anti-trafficking concerns or the controversial effects of some narratives on trafficking. The literature used could be updated with some more recent contributions.

-    In some cases, the article is written in the first person singular, in others in the first person plural. I recommend homogenizing it and, given the nature of the research, it may be appropriate to use the first person singular.

-    I recommend reformatting the article and, in particular, the section titles. Currently, the theoretical framework of the article (The anti-trafficking narrative), is in italics but not in bold and it seems to be a section of the introduction. Similarly, the sections presenting the data (“Financing strategies”, etc.) are presented as if they were sub-sections of the methodological section (“About the method”).

-    The final discussion could benefit from greater dialogue with the critical scientific production on trafficking and emphasize the article's contribution.

Author Response

- Title changed
- Line 74: recent reference added
- Line 102: ídem
- Line 106: fist person singular substituting first plural
- Line 51: Change to impersonal mode
- Line 59: fist person singular substituting first plural
- Line 274: fist person singular substituting first plural
- Line 316: fist person singular substituting first plural
_ Line: 204: New Title added to solve formatting problems
- Line 62: Change cursive to black to solve formmatting problems

Reviewer 2 Report

Comments and Suggestions for Authors

My work shows, for example, more 530 situations of trafficking - very coercive and dangerous - than of trafficking and forced 531 prostitution.

Please explain better this sentence:  My work shows, for example, more 530 situations of trafficking - very coercive and dangerous - than of trafficking and forced prostitution.

Author Response

- LIne 529: Smuggling substituting trafficking (was a translation mistake)